Medical Imaging with Deep Learning 2021

# TG-DGM: Clustering Brain Activity using a Temporal Graph Deep Generative Model

**Simeon Spasov**[*1]                                                SES88@CL.CAM.AC.UK

**Alex Campbell**[*1]                                                AJRC4@CL.CAM.AC.UK

**Giovanna Maria Dimitri** [2]                                GIOVANNA.DIMITRI@UNISI.IT

**Alessandro Di Stefano** [3]                                A.DISTEFANO@TEES.AC.UK

**Franco Scarselli** [2]                                        FRANCO@DIISM.UNISI.IT

**Pietro Liò** [1]                                                PL219@CL.CAM.AC.UK

*Department of Computer Science and Technology, University of Cambridge, United Kingdom* [1]

*Department of Information Engineering and Mathematics, University of Siena, Italy* [2]

*School of Computing, Engineering and Digital Technologies, Department of Computing and Games, Teesside University, United Kingdom* [3]

## Abstract

Spatiotemporal graphs are a natural representation of dynamic brain activity derived from functional magnetic imaging (fMRI) data. Previous works, however, tend to ignore time dynamics of the brain and focus on static graphs. In this paper, we propose a temporal graph deep generative model (TG-DGM) which clusters brain regions into communities that evolve over time. In particular, subject embeddings capture inter-subject variability and its impact on communities using neural networks. We validate our model on the UK Biobank data[1]. Results of up to 0.81 AUC ROC on the task of biological sex classification demonstrate that injecting time dynamics in our model outperforms a static baseline.

**Keywords:** Temporal Graph, Generative Model, Deep Learning, fMRI

## 1. Introduction

The brain can be represented as a temporal graph, where nodes are spatially distributed regions-of-interest (ROIs) defined by a brain atlas. The edges are determined by a measure of dynamic functional connectivity (dFC) applied to the fMRI data. Emerging research suggests that *temporal dynamics of ROI communities* are useful biomarkers for understanding brain function and dysfunction. The majority of existing methods are limited by assuming either static connectivity, or are difficult to scale to many subjects, or are supervised (Ting et al., 2020; Gadgil et al., 2020). Based on these limitations, we propose an *unsupervised* temporal graph deep generative model (TG-DGM) for learning dynamic communities of brain activity from fMRI data. Our model is inspired by Graph Dynamic Embedding (GRADE) (Spasov et al., 2020). In particular, we extend GRADE by introducing multi-graph learning and subject embeddings, giving the ability to quantify subject-specific effects on community membership and dynamics. We demonstrate that our approach learns high-quality representations and that taking into account temporal dynamics improves performance on the task of biological sex classification. Possible applications include using the embeddings to discover novel patient categories, as well as to identify new functional networks (i.e. clusters) of ROIs.

---

[*] Contributed equally

1. The authors would like to thank Dr. Richard A.I. Bethlehem for providing them with the data.

## 2. Method

Denote fMRI data as a set of temporal graphs $\mathcal{D} = \{\mathcal{G}^{(s,t)} : s = 1, \ldots S, t = 1, \ldots T\}$ where each $\mathcal{G}^{(s,t)} = (V, E^{(s,t)})$ represents an undirected graph for subject $s$ at time $t$. Nodes $V = \{1, \ldots, N\}$ remain fixed across time and edges $E^{(s,t)} = \{(w_n^{(s,t)}, c_{n,m}^{(s,t)}) : n, m \in V\}$ evolve dynamically, where $(w_n^{(s,t)}, c_{n,m}^{(s,t)})$ represents the edge that connects node $w_n^{(s,t)}$ to its $m$-th neighbour $c_{n,m}^{(s,t)}$. The edges $E^{(s,t)}$ are produced using a measure of dFC. The goal is to learn subject, community and node embeddings, $\boldsymbol{\alpha}, \boldsymbol{\beta}$ and $\boldsymbol{\phi}$ respectively of size $D$. TG-DGM uses neural networks (NNs) to transform subject embeddings into subject-specific node and community representations. Time dynamics are injected by evolving $\boldsymbol{\phi}$ and $\boldsymbol{\beta}$ with gated recurrent units (GRU). Each node is represented as a mixture of $K$ communities, and each community is a distribution over the $N$ nodes. These distributions are parameterized by NN transformations of the node and community embeddings. For each edge, a community assignment variable $z$ is sampled for the node $w_n$, and then a neighbour $c_{n,m}$ is sampled from the assigned community. The generative process for TG-DGM is:

1. Initialize a *learnable* subject embeddings matrix $\boldsymbol{\alpha} \in \mathbb{R}^{S \times D}$.

2. For subject $s$ in $1 \ldots S$:

   (a) Initialise community embeddings $\boldsymbol{\beta}^{(s,0)} = \text{NN}_\beta(\alpha^{(s)})$, where $\boldsymbol{\beta}^{(s,0)} \in \mathbb{R}^{K \times D}$

   (b) Initialise node embeddings $\boldsymbol{\phi}^{(s,0)} = \text{NN}_\phi(\alpha^{(s)})$, where $\boldsymbol{\phi}^{(s,0)} \in \mathbb{R}^{N \times D}$

   (c) For time t in $1 \ldots T$:

        i. $\beta_k^{(s,t)} = \text{GRU}_\beta(\beta^{(s,0:t-1)})$ for k in $1 \ldots K$, where $\beta_k^{(s,t)} \in \mathbb{R}^D$

        ii. $\phi_n^{(s,t)} = \text{GRU}_\phi(\phi^{(s,0:t-1)})$ for n in $1 \ldots N$, where $\phi_n^{(s,t)} \in \mathbb{R}^D$

        iii. For edge $(w_n^{(s,t)}, c_{n,m}^{(s,t)})$ in $\mathcal{G}^{(s,t)}$:

            A. Produce community mixture coefficients: $\pi_{n,m}^{(s,t)} = \text{softmax}(\text{NN}(\phi_n^{(s,t)}))$

            B. Sample community assignment: $z_{n,m}^{(s,t)} \sim p_{\pi_{n,m}^{(s,t)}}(z|w_n^{(s,t)})$

            C. Produce parameters of node distribution: $\theta_z^{(s,t)} = \text{softmax}(\text{NN}(\beta_z^{(s,t)}))$

            D. Draw linked neighbour: $c_{n,m}^{(s,t)} \sim p_{\theta^{(s,t)}}(c|z_{n,m}^{(s,t)})$

To learn the parameters, we can maximize the probability of the observed data $\mathcal{D}$. The posterior over the community assignment variable $p(z|\mathcal{D})$ is intractable, hence we resort to optimizing the evidence lower bound (ELBO) using variational inference. The ELBO for a single subject $s$ at time point $t$ is:

$$L^{(s,t)} = \sum_{n=1}^N \sum_{m=1}^M \mathbb{E}_{z_{n,m}^{(s,t)} \sim q(z|w_n^{(s,t)}, c_{n,m}^{(s,t)})}[\log p_{\theta_z^{(s,t)}}(c|z_{n,m}^{(s,t)})] - \text{KL}[q(z|w_n^{(s,t)}, c_{n,m}^{(s,t)})||p_{\pi_{n,m}^{(s,t)}}(z|w_n^{(s,t)})]$$

where $M = \text{degree}(w_n)$, $q(z|w_n^{(s,t)}, c_{n,m}^{(s,t)})$ is the variational approximation to the prior over $z_{n,m}^{(s,t)}$, $\mathbb{E}[.]$ is the expectation, and $\text{KL}(.)$ is the Kullback-Leibler divergence. To produce $q(.)$, we simply add the dependency of the posterior over $z$ on the neighbour by augmenting the input to the neural network, which produces the community mixture coefficients during inference. The distributional parameters for $q(.)$ are given by $\text{softmax}(\text{NN}(\phi_{w_n}, \phi_{c_{n,m}}))$. All NNs are implemented as linear layers, and the GRUs have a single hidden layer.

## 3. Experiments & Discussion

We randomly sample fMRI data from $S{=}560$ gender and age (55-75) matched subjects from UK Biobank (Sudlow et al., 2015). The data was preprocessed with standard methods and parcellated into $N{=}360$ ROI time-series of length 490 using the Glasser atlas (Glasser et al., 2016). For each ROI timeseries, correlation based dFC was calculated using non-overlapping window sizes of varying length $W$. The top 1% of correlations were converted into binary edges $E^{(s,t)}$ by thresholding.

Table 1: AUC ROC results on biological sex classification using TG-DGM embeddings

| $W$ | $T$ | $\boldsymbol{\alpha}$ | $\boldsymbol{\beta}$ | $\boldsymbol{\alpha}||\boldsymbol{\beta}$ |
|---|---|---|---|---|
| 490 | 1 | $0.70 \pm 0.03$ | $0.73 \pm 0.04$ | $0.75 \pm 0.03$ |
| 245 | 2 | $0.74 \pm 0.07$ | $0.72 \pm 0.04$ | $0.74 \pm 0.06$ |
| 98 | 5 | $0.78 \pm 0.03$ | $0.78 \pm 0.04$ | $0.80 \pm 0.06$ |
| 49 | 10 | $0.79 \pm 0.04$ | $0.78 \pm 0.05$ | $0.80 \pm 0.04$ |
| 35 | 14 | $0.70 \pm 0.06$ | $0.79 \pm 0.05$ | $0.77 \pm 0.04$ |
| 14 | 35 | $0.80 \pm 0.02$ | $0.79 \pm 0.05$ | $0.81 \pm 0.03$ |
| 7 | 70 | $0.69 \pm 0.06$ | $0.73 \pm 0.07$ | $0.70 \pm 0.06$ |

TG-DGM was trained with the Adam optimizer using a learning rate of $5 \times 10^{-3}$ for 1000 epochs. We set $K{=}7$ and $D{=}32$. We infer the subject $\boldsymbol{\alpha}$, community $\boldsymbol{\beta}$ and node $\boldsymbol{\phi}$ embeddings for all subjects across all time points. To evaluate the quality of the learned embeddings, we use them as inputs to a logistic regression classifier to predict biological sex within a 5-fold cross-validation procedure.

Table 1 presents the AUC ROC obtained from subject embeddings $\boldsymbol{\alpha}$, the time-averaged community embeddings $\boldsymbol{\beta}$ for each subject, as well as their concatenation $\boldsymbol{\alpha}||\boldsymbol{\beta}$ as inputs to the logistic regression. The results demonstrate that at a certain temporal resolution, our model achieves a peak in performance with statistical significance ($p$-values $< 0.05$ when $T{=}35$ against a static baseline $T{=}1$). This shows *dFC has advantages over static connectivity for understanding brain function*. In terms of absolute performance, TG-DGM achieves strong results in sex classification, and unlike supervised methods applied on the same task (Gadgil et al., 2020), our model could be used to *discover novel subject groupings* beyond established categories. TG-DGM is also interpretable as we can associate the community embeddings, and their importance for a given task, to the ROIs.

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
