# OpenReview forum: "TG-DGM: Clustering Brain Activity using a Temporal Graph Deep Generative Model"
_MIDL.io/2021/Conference/Short — MIDL 2021 Poster_

### Official Review · Reviewer_EbM9 · 2021-04-25

**Confidence:** 5
**Final Rating:** 2

**Summary:**

The authors propose a graph deep generative model (coined TG-DGM) to perform community detection to obtain a time-evolving clustering of brain regions in an unsupervised learning setup. The authors demonstrate that their algorithm learns discriminative embeddings suitable for biological sex classification, as validated on the UK Biobank data, and outperform a static version of their model, inspired by their previous paper (GRADE, Spasov et al 2020)

**Strengths:**

1. For the most, the manuscript is really well written, with the model formulation, inference, and experimental paradigm explained clearly and precisely. For a short submission, the experiments comparisons are comprehensive, well-motivated and substantive for the thesis statement of the paper.

2. The method provides tangible performance gains over the static version of the model, strengthening the argument for utilizing dynamic over static rs-fMRI connectivity.

**Weaknesses:**

The following points need clarification:

1. I am not sure what the authors mean by a linear regression classifier since the target variable (biological sex) is categorical.

2. How is the dFC measured in this case (Pearson's correlation coefficient?), is it thresholded to convert into binary edges?

3. "First, the results demonstrate that temporal dynamics increase performance with statistical significance (p-values < 0.05 when T=35 against a static baseline T=1). " This statement reads as if the rest of the dFC results are not statistically significant. If this is indeed the case, I would argue that strictly speaking, W (window length) is a tunable hyperparameter that needs to be set using a separate validation set

4. In light of point 3, how do the authors determine parameters such as learning rate, epochs, K, D etc?



**Deanonymize Review:**

no

**Detailed Comments:**

1. Could the authors please explain the setup of the linear regression classifier?

2. Could the authors please explain the hyperparameter selection strategy?

3.  "Second, our results are comparable in absolute performance to supervised graph convolutional methods (Gadgil et al., 2020) on the same task. This means TG-DGM learns high-quality embeddings, and unlike supervised methods could be used to discover novel subject groupings beyond established categories."

This statement is a strange comparison to make considering that (Gadgil et. al, 2020) does examine the same task, but on HCP/NCANDA data and not on UK Biobank. While I do see the difference between the supervised vs unsupervised paradigm, I still think its a bit misleading as presented.



**Justification Of The Rating:**

It seems to me that the main contribution of the work is methodological, and in this regard I appreciate the contribution. Nevertheless, the points detailed in weaknesses and comments need to be addressed sufficiently, which is why I currently assign a borderline score.

**Paper Type:**

methodological development

**Special Issue:**

no

---

### Official Review · Reviewer_a67Q · 2021-04-27

**Confidence:** 4
**Final Rating:** 4

**Summary:**

The authors propose a temporal graph deep generative model (TG-DGM)  to identify communities of brain activity using fMRI data. Their proposed model builds on top of the GRADE model, by adding subject embeddings, capturing inter-subject variability. To test their model, the authors used embeddings of 560 subjects, generated across several time points using the TG-DGM in an unsupervised fashion, as input to a linear regression classifier to classify sex. They report results comparable to the literature in which a supervised approach was used.

**Strengths:**

The paper is well written, the motivation is clearly explained and the methods are detailed. The experiment is relevant and nicely shows how the time dynamics affect the results. The interpretable component of the model is very interesting to be investigated in future work.

**Weaknesses:**

The following are more suggestions than weaknesses as these additional information would be difficult to add given the page limit:

It would be nice to have a direct comparison to Gadgil et al in the result table.

We can see that the AUC ROC results increase and then decrease while T increases, but it is not discussed.

Details on the choice for hyperparameters could be added, for instance K=7 and D=32, was it determined empirically?

**Deanonymize Review:**

no

**Detailed Comments:**

In Table 1 line 2, W should be equal to 245.

**Justification Of The Rating:**

The authors proposed a novel approach and gave a detailed explanation of the extension they made based on previous work. The method was tested using a large sample size and on a task that is relevant and easily comparable to previously published work. The proposed method gives promising results.

**Paper Type:**

both

**Special Issue:**

no

---

### Meta-Review · Area_Chair_p1eC · 2021-05-09

**Recommendation:** Accept (Poster)
**Confidence:** 4

**Metareview:**

Both reviewers liked the methodological contribution and while both also highlight a number of points that are currently unclear a thorough revision for the final version should make this into a good MIDL short paper. Again we would like to encourage the authors to publish a link to public source code, and do not understand why this is disadvantage for an upcoming journal submission (the contrary is true).

---

### Decision · Program_Chairs · 2021-05-11

Accept (Poster)